# Researchers and Their Experimental Models: A Pilot Survey in the Context of the European Union Health and Life Science Research

**DOI:** 10.3390/ani12202778

**Published:** 2022-10-14

**Authors:** Lorenzo Del Pace, Laura Viviani, Marco Straccia

**Affiliations:** 1Independent Consultant in Philosophy of Science, 04100 Latina, Italy; 2Independent Consultant in Science Ethics, 4053 Basel, Switzerland; 3FRESCI by Science & Strategy SL, 08233 Barcelona, Spain

**Keywords:** Directive 2010/63/EU, animal experimentation, NAM, new approach methodologies, experimental methods, experimental models, phasing-out, science policy, researcher view, European policy, survey, biomedical research

## Abstract

**Simple Summary:**

Scientists in biomedical research use models and methods to constantly improve health in society. This research heavily relies on animal experimentation, and in recent decades, research and researchers have been questioned by societal stakeholders about their way of conducting research. In order to inform science policy makers, we asked the researchers about the use of their experimental models and their view about the role of external stakeholders in their work.

**Abstract:**

A significant debate is ongoing on the effectiveness of animal experimentation, due to the increasing reports of failure in the translation of results from preclinical animal experiments to human patients. Scientific, ethical, social and economic considerations linked to the use of animals raise concerns in a variety of societal contributors (regulators, policy makers, non-governmental organisations, industry, etc.). The aim of this study was to record researchers’ voices about their vision on this science evolution, to reconstruct as truthful as possible an image of the reality of health and life science research, by using a key instrument in the hands of the researcher: the experimental models. Hence, we surveyed European-based health and life sciences researchers, to reconstruct and decipher the varying orientations and opinions of this community over these large transformations. In the interest of advancing the public debate and more accurately guide the policy of research, it is important that policy makers, society, scientists and all stakeholders (1) mature as comprehensive as possible an understanding of the researchers’ perspectives on the selection and establishment of the experimental models, and (2) that researchers publicly share the research community opinions regarding the external factors influencing their professional work. Our results highlighted a general homogeneity of answers from the 117 respondents. However, some discrepancies on specific key issues and topics were registered in the subgroups. These recorded divergent views might prove useful to policy makers and regulators to calibrate their agenda and shape the future of the European health and life science research. Overall, the results of this pilot study highlight the need of a continuous, open and broad discussion between researchers and science policy stakeholders.

## 1. Introduction

In September 2021, the European Parliament adopted a resolution with 667 votes to 4 on an EU-wide action plan for phasing out the use of animals in research and testing [1]. Members of the European Parliament requested sufficient medium- and long-term funding and coordination to further promote the development and deployment of alternative methods and models [1].

The current EU legislation providing the legal framework for animal use in science is Directive 2010/63/EU [2], which dictates the animal welfare terms to follow when using animals for research purposes. The Directive makes mandatory the application of the Three Rs Principles (replacement, reduction and refinement) of animal use in science, with the final aim of Directive 2010/63/EU being the ultimate replacement of procedures on live animals. The EU willingness in animal welfare has brought, so far, a stop to animal testing for finished cosmetics products, since 11 September 2004, and for cosmetic ingredients, since 11 March 2009 [3].

The statistics on the use of animals for research are monitored and made publicly available by the European Commission, through the recently launched online platform ALURES [4], which provides a new high level of transparency on animal-based research. In fact, Directive 2010/63/EU, amended by Regulation (EU)2019/1010 (Article 6) [5], requires the European Commission and the EU Member States to make publicly available the statistical data on the use of animals for scientific purposes. The latest statistical summary reports [6] available on the use of animals for experimentation, published by the European Commission for 2018, reported the use of more than 10 million animals (10,572,305), in the EU-28 and Norway [7].

Interestingly, most of the animals (73%) were used in basic (46%) and translational-applied (27%) research. In particular, for studying human diseases and non-regulatory toxicology, falling into the category of translational and applied research, a total of 1,873,677 (18%) animals were used in 2018. Although animal experimentation has led and still leads to successful medical treatment for people in our society, the latter data on animal use in research are linked to an increasing clinical trials failure rate [8]. This augmented failure is also due to the poor translatability from animal preclinical models to human patients [9,10,11] raising questions concerning the ethical issues of animal use in, and the economic cost of biomedical research [9].

EU legislation together with the societal pressure to stop animal experimentation [12,13] and the data from the statistics are generating an important external pressure on researchers of health and life sciences using animals [14,15].

The existence of a variety of stakeholders continuously engaged in favour [16,17] or not in accord with [18,19] animal research activities, including both researchers and citizens, shows what could be the epiphenomenon of a certain degree of underlying societal polarization on the theme [20]. A polarization that through civilized dissent and scientific argumentation eventually reaches the political sphere, raising the question of what influence it produces on the scientific community.

In fact, the debate has widened to other stakeholder communities: scientists and regulators have been joined by a host of committed societal interlocutors [12,21]. These are not directly involved in the practice of science, but they have great interest in it. These new societal interlocutors have never before experienced the capability to successfully engage decision makers on scientific topics, raise public support, run information and awareness campaigns, and succeed in enacting boundaries to scientific research through policymaking. This golden era of science, born from the effective deployment of multiple solutions to overcome the COVID-19 pandemic, coincides with the age of full accountability, where responsibility towards patients, citizens and granting institutions is just part of the answers that scientists are called on to explain their work.

Both the research community [22,23] and those stakeholders (policy makers, non-governmental organisations, industry, funding bodies and regulators) which are fully involved in the Three Rs debate are influenced and react to this polarization, something that has prompted the European Commission to the analysis of the drivers and barriers to the improvement of cross-disciplinary solutions in the biosciences [24]. Many efforts [25] are ongoing to promote and inform all parties of the policy development activity [26], the regulatory needs [27] and to appropriately relay to the scientific community the calls from non-scientific stakeholders [12].

The aim of this study was to record researchers’ voices about their vision on this ongoing scientific transformation, to reconstruct as truthful as possible an image of the reality of health and life science research, by using a key instrument in the hands of the researcher: the experimental models. In this view, it is important for society and for scientists (1) to understand researchers perspectives on the selection and establishment of their models of use and (2) to share the researchers community opinion regarding the external factors and stakeholders influencing their professional work.

In the interest of advancing the public debate and inform research policies, the results of this survey will provide reliable data to the stakeholders, as a starting point to mitigate public perception biases regarding the European research environment.

## 2. Material and Methods

From October 2020 to June 2021, a survey was conducted among health and life science researchers working in Europe, in order to collect their views on the use of the experimental models in research. An online questionnaire [28] was developed and distributed among a conspicuous number of European researchers.

### 2.1. Questionnaire Design

The outline of the questionnaire was developed by the three authors and was based on: (I) a previously conducted survey among researchers involved in biomedical research in European bioclusters ecosystem and (II) multiple rounds of peer reviews through researchers and experts. The survey was anonymous, descriptive in nature and included both qualitative and quantitative questions. The questionnaire was tested by 10 researchers and adjusted on the basis of their feedback. The use of closed-ended questions ensured that respondents were consistent in their answers. There was room to give additional comments to questions in case a respondent did not consider the provided set of answers exhaustive. Some questions allowed for multiple answers, e.g., on information sources. The language of the original questionnaire was English.

The survey contained 22 questions grouped into 8 thematic areas: Participant information (9), Research Models (4), Funding (2), Drivers and Barriers (3), Education (1), 2010/63/EU Directive (1), and Future Perspectives (2). Of those 22 questions, 5 were “container questions”: respondents were not asked to submit a single answer, rather they were confronted with either rating a series of statements related to a specific topic or issue, or to express their opinion on a topic, or to assess concerns and/or expected benefits specific to certain topics. Container questions are tools to investigate the same specific issue, attitude, concern, or acquired knowledge from different viewpoints. This brings the gross total of survey answers to 49 for researchers with no teaching activity and to 52 to researchers involved in teaching activities.

### 2.2. Questionnaire Distribution

A link to the online questionnaire was distributed among private companies (contract research organisations, pharmaceutical and chemical industry, etc.), major universities and research institutions, European scientific societies and associations through several actions: (1) e-mail campaign: e-mails were sent to subjects of each European country, EU member states plus Switzerland and UK. Most of the emails were sent to specific recipients identified as possible point of contacts, and to a minor number of institutional addresses, with a final total number of 16,668 e-mails; (2) LinkedIn post: the survey flyer and a direct link to the survey was posted on the LinkedIn profile of FRESCI and its consultants with 43 specific hashtags, tagging 20 direct contacts from key European associations, with a total of 1143 views.

The questionnaire flyer was also posted in 11 LinkedIn groups: JRC Summer School (154 subscribers), Altertox Academy (269 subscribers), WC11Maastricht (442 subscribers), ESTIV (455 subscribers), Preclinical Toxicology Consultation Network (5491 subscribers), Stem cell and Regenerative Medicine (6923 subscribers), Neurodegenerative disorders (7710 subscribers), FENS (12,808 subscribers), Science communication dissemination and exploitation (12,842), Stem Cell Research (14,064 subscribers), Science Network (281,842 subscribers); (3) other social media channels: Facebook, Instagram and Twitter were also used to distribute the survey link and information; (4) Hub strategy: relevant senior researchers or department heads were directly contacted, and 5 of them shared the survey through their official communication channels; (5) Congress and meeting presentation: the survey was presented to many congresses and meetings, such as Building Bridges Champalimaud Workshop, 11th World Congress on Alternatives and Animal Use in the Life Sciences 2021, and JRC summer school 2021, among others.

### 2.3. Data Analysis

Despite the efforts of the authors, some misinterpretations cannot be ruled out. To safeguard anonymity and to exclude potential bias, the survey data were disconnected from the respondents’ backgrounds and contact details. The results were analysed both per question and inter-related to perform segmentation analyses. The closed-ended multiple answer questions, the Yes/No questions, and the questions with scaled answers were analysed through counting frequencies in Excel. The answers to open questions were listed and categorized by inductive analysis. The data were analysed by the last author (MS) and discussed with LDP and LV.

The total number of participants was 119; However, two contributors participated twice, and only the most complete set of answers was considered. One contribution was representing the common opinion of a department with sixty researchers, as stated by the contributor in the final comment section. However, we still considered this contribution as a single data point. Some of the participants did not answer all the proposed questions. As such, the sum of results in the presented charts will not always correspond to the total number of participants. Container questions are graphed in a percentage scale to normalise the weight of the different groups; in addition, the absolute number of voters is also reported in the respective histograms.

### 2.4. Limits of the Study

The number of participants (117) has no statistical representation of the whole health and life science researchers population in Europe; however, this survey had comparable participation in other international surveys in the Three Rs field or among researchers (see Appendix A).

Since the representation of Netherlands participants was half of the sampled population, we checked the unwanted sampling bias. We tested the two populations for the equality of variance (Test F) and for the unpaired two-tailed Student *t*-test for the following objective characteristics: user model type; Euraxess researcher descriptors [29]; involvement in regulatory science projects, engagement in teaching activities, use of genetic engineered models, opinion regarding high-impact factor journals publication and business exploitation. Test F found variance difference (*p* = 0.032) in the answer distribution regarding the opinion on high-impact factor journals publication and business exploitation. However, no statistical difference among the two populations’ set of answers were observed through the Student’s *t*-test.

## 3. Results

### 3.1. Participants Information

From 27 October 2020 to 31 August 2021, we recorded a total of 119 contributions to our survey, 117 of which were unique replies. The survey participants were almost equally distributed by gender, with 59 female and 56 male respondents (Figure 1). Researchers in the age range of 35–44 years old (y.o.) were the most represented, 37 in total. Female researchers below 35 y.o. have contributed considerably more than male researchers of the same age, 21 versus 9 contributions. The opposite is true for the over 44 y.o. participants group, where male respondents numbered higher than the female counterparts, 31 versus 19 (Figure 1).

Participants conducted their research in Europe, as targeted, with a major representation from the Netherlands (57), followed by Spain (14), Italy (12), France (6), Germany (6), Switzerland (4), Denmark (3), Portugal (3), Greece (2), United Kingdom (2), Poland (2), Austria (1), Belgium (1), Ireland (1), Lithuania (1) and Romania (1) (Appendix A).

The majority of respondents have a single affiliation (79.5%), with academia being the most represented (72.7%), followed by governmental research institutions (13.7%), non-governmental laboratories (6.0%), foundations (1.7%), biotech companies (1.7%) and regulatory bodies (1.7%). Academics were the group with more participants having two or more affiliations (17.1%), followed by non-governmental (2.6%) and governmental (0.9%) researchers.

Researchers were classified by using the Euraxess research profiles descriptors [29], where R1 is up to the point of PhD; R2 are PhD holders or equivalent who are not yet fully independent; R3 are researchers who have developed a level of independence; and R4 are researchers leading their research area or field. Of those who responded to the survey, the majority of researchers had an R3 profile (32.0%) followed by R2 (27.0%), R4 (22.0%) and R1 researchers (19.0%).

### 3.2. Participants Research Fields and Models

The most common fields of research reported by participants were in the neuroscience sector (15.6%), followed by toxicology (9.9%), oncology (5.7%), biomedicine (5.0%) and regenerative medicine (4.3%). Another 49 disciplines were also registered (see Appendix A).

We then classified the participants by the type of model they were mainly using, considering if they were using models based on material or data proceeding from animals or humans. A total of 37.6% of the participants relied on both human and animal models in their experimental research, followed by 33.3% of researchers using human-based models and 25.6% of respondents implemented animal models. Moreover, 3.4% stated a reliance on other methods, without indicating the data/samples source. Furthermore, of those who responded (98), forty-one were using genetic engineered models.

Deepening the analysis on the models used by the participants, the most frequent model used by participants were primary cell cultures (60), followed by immortalized cells (53), biochemical assays (45), stem cells (45), live organism (33), organoids (33), biopsies (23) and other microphysiological systems (21; i.e., Organ-on-Chip). Other models were also reported.

### 3.3. Experimental Model Establishment

Participants were asked whether their model of choice was selected or established. The majority of answers pointed to literature review (58). Fifty-one researchers also answered that the model was self-developed, which is particularly true for human-based model users (23) and researchers who used both animal- and human-based models (22) (Figure 2). Inheritance represented the third most frequent factor for models selection: 37 respondents declared that the model in use was already established in the research group of work. Additionally, the mentor experience was reported as a factor to select the experimental model. Lastly, benchmarking is the fourth method most reported to select or to establish a model, especially for those researchers using both human- and animal-based models. Five researchers stated their chosen model to be the only one available, while seven reported to having relied on other methods to choose and select their models.

Of those respondents who reported that the model in use was their own development, nineteen of them did not indicate reliance on any other method to establish their models. The distribution of these 19 respondents by their researchers profile is the following: 12 for R3–R4, 3 for R2–R3 and 4 for R1.

### 3.4. The Model in Use

We asked the participants to rate a series of statements regarding their experimental models. The rating ranged from one to five stars, with one star indicating complete disagreement and five stars full agreement with the statement. We stratified the researchers’ answers by the type of models they used: animal-based, human-based and both animal- and human-based models (Figure 3).

*The model I am using has been standardized/qualified or regulatory validated:* Users of animal models stated that their models are standardized and/or qualified or subject to regulatory validation to a certain extent, and they are, in any case, in major proportion compared to human-based model users. Users of both type of models rated the statement with a more balanced distribution with a tendency to consider their models as standardized/qualified or subject to regulatory validation.

*The model I am using is affordable for my organisation, and/or I have access to good structures to use it:* All three groups did show high agreement with the statement, especially in the case of users of animal-based models.

*I have access to results of previous studies that used the same model that I am using:* Human-based model users and users of both types of model reported greater difficulty in accessing previous studies’ results in comparison to animal model users.

*The model I am using was part of my educational curriculum:* Notably, most of the respondents reported that their models were not part of their educational curricula. Just a small proportion of respondents considered the models they employed as part of their educational path, in particular for those researchers using human-based models.

*I collaborate with other groups and researchers using or further developing the same experimental model I am working with:* All 3 types of users reported collaborating with peers to further develop the used models, highlighting the importance of the research network and collaboration. Partial disagreement was recorded in the case of animal-based model users.

*I know of other models relevant to my work, but I don’t have access to them:* The majority of users, independently of model choice, disagreed with the question, meaning that only a small minority of them find themselves in the situation of knowing alternative methods, but not having the material opportunity to use them in their work.

### 3.5. Perspectives on Model Acceptance

To investigate the importance for researchers regarding the use of accepted models, we asked the following question: “How important is it for you that your chosen model is widespread and established in the scientific community at large?” (Figure 4).

An overwhelming majority of users deemed it either important (47.9%) or very important (38.5%) that their chosen model is widespread and established. Only a small minority (5.1%) seemed to consider the status of their chosen model in the research community as non-discriminant for their choice.

After surveying respondents on the importance they attach to the community status of a research model, further questions were posed to analyse the reaction to outlier cases. Specifically, we probed the reaction to the eventual need to rely on not-yet-peer-reviewed methods, and the associated concerns or benefits expected by the participants by asking the following question: “If you had the need to use a still-non-peer-reviewed model or method, how would you rate the following concerns and benefits?”.

*Difficulty in using or accessing new equipment:* On the one hand, 38.1% of respondents were not preoccupied with the possibility that the use of a non-peer-reviewed model or method might hinder access to new equipment. On the other hand, 31.9% of researchers stated that when using a non-peer-reviewed model it can be difficult to access or use new equipment. Finally, 30.0% considered the use of non-peer-reviewed models neither a benefit nor a concern in accessing or use new equipment, especially for users of both animal- and human-based models.

*Chance to access new funding opportunities (different funding programs, different donor organisations, etc.):* Opinions seemed to be almost equally split between the three possibilities of increased/neutral/decreased chances of new funding opportunities, with a similar trend for the three user groups. Neutral outlooks and negative outlooks were identical (34.5%), with just slightly lower numbers sharing a positive outlook (31%). Disagreement was especially registered among users of animal-based and both animal- and human-based models, while the higher agreement was registered among human-based model users.

*Increased probability of breakthrough solutions:* The majority of respondents (44.2%) considered that the use of non-peer-reviewed models would increase the chance of breakthrough solutions. About a third expressed a more neutral attitude (36.3%), while only a minority disagreed (19.5%).

*Difficulty in engaging superiors/supervisors:* Over half of the respondents (51.3%) did not consider the use of non-peer-reviewed models a barrier to superiors and/or supervisors engagement. Only 22.1% reported that it could represent a difficulty. In the specific case of animal-based model users, the use of a non-peer-reviewed model was completely balanced with respect to the difficulty or not in engaging superiors and/or supervisors.

*Difficulty in publishing your research:* Animal-based model users tended more (53.3%) to consider that the implementation of non-peer-reviewed models in their research would increment the difficulty in publishing their results. For human-based model users, the opinion was more balanced, with a slight tendency (41.0% vs. 35.9%) to not see any difficulty in publishing their research when using non-peer-reviewed models. In the case of users of both types of model, the 50% disagreed with the difficulty of publishing their research compared to the 36.4% who considered the use of non-peer-reviewed models a barrier for publishing.

*Difficulty in comparing results among models:* The majority of respondents (60.2%), independently of their model of use, considered that the use of still-non-peer-reviewed models is an obstacle to compare results among models. Sixteen (13.2%) respondents from the three user groups expressed disagreement about facing such a difficulty when comparing results from non-peer-reviewed models. Interestingly, three users (2.5%) of both types of model were in full disagreement with this difficulty. The remaining 24.1% of the respondents answered neutrally.

### 3.6. Willingness in Methods and Model Sharing

Researchers were also sampled for their willingness to take part to a widened collaborative process, in this case exemplified by a plausible shared, open data platform on which to collectively connect to share knowledge on new methods and models. Hence, we asked: “Would you consider to connect on a shared open-data virtual platform with peers to develop, characterise, validate and share knowledge on new methods and/or models in your research field?”.

The overwhelming majority answered positively, reinforcing the commonly held conception that research thrives through reliable communication channels among researchers, essential tools to propel our knowledge forward in health and life sciences. Participants were asked about their willingness to connect with their peers through a shared open-data virtual platform and the great majority (84.0%) answered positively. Only 16% were negative on this proposal.

### 3.7. Research Models Drivers

The investigation on the perception of models, and on practical experience with them, was continued through questioning the respondents on the need of having animal experiments to complement in vitro or in silico human-based experimental models research, in order to publish in high impact factor journals or to foster business exploitation.

Complementing in vitro/in silico human-based results with animal experiments was seen as a necessity by the majority of respondents (58.1%) to publish in high impact factor journals or for the economic exploitation of their research.

Users of animal-based models were more strongly convinced of the necessity of complementing experiments (73.3% vs. 26.7%), and a similar, albeit less marked trend was evident for users of both animal- and human-based models (59.1% vs. 40.9%). Human-based model users are the only group with a majority convinced of the absence of such necessity (53.8% vs. 46.2%) (Figure 5).

To those respondents, who answered regarding the need of complementing in vitro/in silico human-based results with animal experiments to publish in high impact factor journal or for business exploitation, we also asked for their direct experience (Figure 6). Notably the majority of all 3 groups had a direct experience on the issue, reinforcing their belief (Figure 6).

### 3.8. Research Funding Opportunities Perception

Through the agreement level scoring of another series of statements, again rated from 1 to 5 stars, an analysis was performed of the personal expectations towards the evolution of future funding opportunities and to the level of future funding and project scrutiny (Figure 7).

*I am positive I will have increased funding opportunities in my research field:* The total distribution of researchers response was almost equally divided on this question, only sceptics were slightly more represented (29.2% optimistic vs. 31.0% neutral vs. 39.8% sceptic). Disaggregating the results, human-based model users were equally distributed on this topic (30.8% optimistic vs. 35.9% neutral vs. 33.3% sceptic) in comparison with the other two groups. In fact, animal-based model users answers were more polarized (36.7% optimistic vs. 20.0% neutral vs. 43.3% sceptic). In contrast with the other two groups, the answers by the users of both types of model showed less optimism in an increase in funding in their research field (22.7% optimistic vs. 34.1% neutral vs. 43.2% sceptic).

*The applications and limitations of research methods/models will be further scrutinised in grant applications:* Users of both human- and animal-based models were convinced that applications and limitations of research models will be further scrutinised (54.5% vs. 11.4%). The situation was relatively more balanced for human-based (43.6% vs. 35.9%) and animal-based models users (36.7% vs. 26.7%), where both opposite perceptions are almost equally represented, with a small advantage for those believing in further scrutiny.

*More national funding will be directed to non-animal methods in the mid-term*: The majority of respondents (56.6%) expressed conviction that more funding for non-animal methods at national level will increase in the mid-term. Proportionally, more convinced in such mid-term strategy were the animal-based models users (63.3%) than the users of both animal- and human-based models (59.1%) or the users of human-based models (48.7%). About one-fourth of the latter two groups expressed a neutral opinion.

*More national funding will be directed to animal-based methods in the mid-term:* Interestingly, the vast majority of respondents considered an increase in the mid-term funding towards animal-based methods unlikely. Strong agreement of a funding increase at national level was expressed only by three animal-based model users.

*Calls for projects will increasingly incentivise integration of different disciplines and methods:* Most of the respondents (74.3%) showed partial or full agreement with the proposed statement. A neutral opinion was held by 14.2% of respondents, while those in partial or complete disagreement were 11.5%.

*Calls for projects will increasingly incentivise strategies to translate research results into societal impact:* A great majority of all three groups of model users considered this statement true (48.7% are in full agreement, and 29.2% in partial agreement), believing that a call for projects will increasingly stimulate the translatability of research results to boost societal impact.

### 3.9. Experimental Research Teaching Experience

Teaching reflects the way a model, and a scientific mindset, are passed from one generation of researchers to the next. Hence, surveying what teaching media are employed, how they are chosen and what experimental works presented has great relevance in understanding what kind of research perspectives are passed along to further generations. Of the 117 survey participants, 85 (72.6%) were engaged in teaching activities within their organisations (Figure 8). They were asked whether, in the course of those teaching activities, they were presenting mostly human-based or animal-based experimental works. Animal-based model users mostly (65.0%) declared using experimental works of both kinds. The same, although to a lesser degree (58.3%), was true for users of both animal-based and human-based models. A diametrically opposite answer came instead from users of human-based models, where a very large majority (79.2%) chose human-based experimental works for teaching (Figure 8).

We also asked the participants for the type of teaching media used and the ones they preferred.

The most common teaching media appeared to be in vitro models, with 42 overall users, and 6 for which it was also the preferred. Screen-based simulators were used by 17 respondents and preferred by 6. Virtual reality was also reported as used by 11 and preferred by 7; this was followed by 15 voters of other in silico methods, which was preferred by 3 of those respondents. Full-body simulators, anatomical parts models, live animals, cadavers and actor-based simulations were also reported as teaching media with a smaller number of preferences, with the exception of the full body simulator that was the preferred by seven respondents.

Another important information is regarding the establishment of teaching media, so we asked: “How did you choose or establish your main teaching tool/media?”.

Overall, 43 (37.7%) respondents declared the used teaching media was their own development, whereas 24 (21.1%) stated that it was already established in their field and/or department. A total of 18 contributors (15.9%) established their teaching media and/or tools based on peers’ experience, 12 respondents (10.5%) reviewed the literature to search for educational resources, and 12 (10.5%) compared different tools and media. Only 5 (4.3%) respondents declared that the tools/media they use were the only ones available (Appendix A).

### 3.10. Continuing Education in Life Science and Biomedical Research

Continuing education and constant training are essential facets of the career of a researcher. The evolution and dissemination of knowledge often create the conditions to change established habits in specific research fields, and to influence and expand the wider perspective of the researchers.

Contributors were asked regarding their attendance to workshop or courses. We segmented their answers by the Euraxess researchers profiles (Figure 9) and by the type of model they used (Figure 10).

The group reporting higher continuing education rates was the R2 (87.7% of the R2 group total answers), closely followed by R3 (82.8% of the R3 group total answers), R4 (82.2% of the R4 group total answers) and R1 (82.1% of the R1 group total answers). R2 and R3 researchers particularly reported higher participation in animal welfare and non-animal models workshops, at 57.9% and 60.3%, respectively (Figure 9). Participation in improving experimental design workshops was also reported by all four groups of researchers, with an attendance between 17.2% and 28.2%, depending on the respondents group. Subgroups ranging from 2.2% to 5.3% of the different profiles of researchers declared to have attended other types of courses or workshops. Notably, between 12.3% and 17.9% of each subgroup of researchers declared not having attended any workshop or course (Figure 9).

When segmenting the results by the type of experimental model used by the respondent, the highest level of attendance has been recorded by the users of animal-based methods (93.3%) (Figure 10). Users of both animal- and human-based models followed relatively closely (85.7%). Last in terms of attendance was the human-based model users group (68.8%). Human-based model users reported higher attendance in non-animal models course (33.3%) compared to the other groups (21.7%, animal-based models users; 21.4%, both model users) but lower attendance in experimental design courses (14.6% versus 28.3% and 25.0% of animal-based model users and both model users, respectively) (Figure 10).

### 3.11. Perception of Stakeholder Participation in Science

Research policy is shaped and agreed through the interaction of a multiplicity of stakeholders, with scientists representing just one of those groups. However, science policy has immediate repercussions on the everyday professional activities—and possibilities—of the researchers. Their appraisal of the participation of stakeholders not directly involved in research activities in the definition of scientific policy is then of great significance.

Overall, respondents showed a consensus on the potential usefulness of the participation of most of the stakeholders in the discourse orienting research. The major exception was represented by political parties, valued by 49.6% of respondents as either as potentially dangerous or dangerous. The most negative perception was expressed by users of animal-based models, where 30.3% of respondents considered them dangerous (Figure 11). On the other side of the spectrum, most of the researchers considered that the participation of research foundations in the debate was useful (72.6%) (Figure 11).

All participants considered in general animal research non-governmental organisations (NGOs) rather useful, although with a certain hesitation, especially in the case of human-based model users. Additionally, animal protection NGOs registered a general approval, although 30% of the users of animal-based models, and 20.4% of the users of both animal- and human-based models expressed neutrality (Figure 11).

The opinion on other research NGOs showed higher polarization: while 74.3% of human-based models users expressed a positive perception, a majority of users of animal-based models (56.6%) expressed instead a negative perception; users of both animal- and human-based models were almost equally split, with a significant minority (45.4%) manifesting apprehension (Figure 11).

Patients associations were seen largely useful by all three types of model users, however some users of each subgroup (from a 20.5% to a 30%) reported perceiving the participation of this stakeholder segment as potentially dangerous (Figure 11).

Regulators were also seen very useful by all three types of users. Some users reported a potentially dangerous participation of this stakeholder segment (27.3% of both types of model users, 33.3% of animal-based models and 7.7% of human-based models users). A similar trend was observed in the case of participation of policy makers to the science policy debate. Interestingly, 9.1% of users of both types of model considered policy makers as a danger (Figure 11).

Pharmaceutical associations were favourably perceived by all the groups (60.2%), while other industry associations received a more polarized response (47.8%, potentially useful/useful; 10.6%, neutral; 41.6%, potentially dangerous/dangerous).

### 3.12. Researchers Considerations on Directive 2010/63/EU

Directive 2010/63/EU is the regulatory framework regulating the use and selection of experimental models for the entire European Union. One the one hand, the Directive strives to protect animals used for research and other scientific purposes by establishing the necessary conditions for their welfare, while also calling for a continuous endeavour for the replacement and refinement of animal-based methods. On the other hand, the Directive 2010/63/EU also determines and causes an evolution in scientific practice through models and methods selection and use, since it envisions the complete replacement of animals for experimentation purposes as soon as it is scientifically possible to do so. It is the cornerstone legislation regulating and indirectly establishing a vision for European health and life science research. Given its importance, we decided to test participants, to evince what was their general level of knowledge and their effective comprehension of the Directive’s content.

The intentional convoluted question was necessary to discern a very superficial understanding of the Directive 2010/63/EU (i.e., animal protection legislation) from a more structured comprehension of its existing nuances. We segmented their answers by the Euraxess researchers profiles (Table 1) and by type of model users (Table 2).

Of the 117 survey participants, 58 answered by saying they did not know whether the Directive 2010/63/EU adequately acknowledges the animal model by virtue of it being well known, predictable, widely used and established. Twenty-seven felt the Directive does not adequately acknowledge animal models because it requires the use of alternative methods and techniques if available. Seventeen participants considered that the Directive acknowledges the currently irreplaceable value of animal research. Additionally, 15 considered that the Directive poorly acknowledges the animals models as it aims at phasing out animal research.

Segmenting the respondents by their Euraxess research profiles, we observed that most of R1 researchers (63.6%) did not know how to answer the question, followed by R3 (55.3%), R4 (42.3%) and R2 (38.7%). Notably, no single R1 researcher answered that the Directive poorly recognises the value of animal models (Table 1), whereas R2 and R3 researchers answering that the Directive poorly acknowledges the animal models’ virtue because its aim is to phase out animal research were more than the double of their R4 counterpart (R2 = 19.4% and R3 = 18.4% versus R4 = 7.7%) (Table 1).

A total of 26.9% of R4 researchers considered that the Directive 2010/63/EU significantly recognises the irreplaceable value of animal research, compared to 12.9% of R2, 10.7% of R1 and to 7.9% of R3 researchers (Table 1). The percentage of R2 researchers considering that the Directive is not acknowledging the value of animal models was 29.0%, whereas for R1, R3 and R4 researchers it was 22.7%, 18.4% and 23.1%, respectively (Table 1).

The segmentation by type of users showed that more than half of animal-based model users (53.3%) or users of both animal- and human-based models (52.3%) users did not know how to answer the question (Table 2). This is a higher percentage than that registered for human-based model users (43.6%).

On the one hand, 30.8% of human-based model users reported that the Directive 2010/63/EU requires the use of alternative methods and techniques if they are available, compared to 20.4% of both model types users and to 16.7% of animal-based model users (Table 2). Moreover, 18.2% of both types of model users also answered that the Directive acknowledges the irreplaceable value of animal research, while human- and animal-based model users were less in agreement with this answer, at 12.8% and 10.0%, respectively (Table 2). On the other hand, 20.0% of animal-based model users considered that the Directive 2010/63/EU poorly recognises the value of animal models, since its final mission is to phase out animal research. This answer was shared by 12.8% of human-based and by 9.1% of both models users (Table 2).

As 59 out of 117 respondents were involved in regulatory science projects, we also segmented the answers regarding the Directive 2010/63/EU based on their involvement in a regulatory science project. Responses were similar among the two groups of researchers through this analysis (Appendix A).

### 3.13. Researchers’ Conditions for Animal-Based Research Phased Out

The Directive 2010/63/EU acknowledges the ultimate goal is to move European health and life science research towards a new era not relying on animals experimentation. While not achievable in the short term, this is a clear goal towards which scientific policy will work by deciding future funding and project evaluation guidelines. Surveying respondents on the conditions for a phasing out of animal research is, thus, essential to obtain their views of the junction where European research is, and what conditions have been already, or ought to be met before such an advance can take place. Seventy-five respondents (64.1%), from the three groups of model users, either fully or partially agreed that a knowledge gap must be filled before phasing out the animal experimentation (Figure 12). However, eleven human-based model users (28.2%) felt neutral in this regard, whereas 16.2% of respondents either partially or fully disagreed, and the lead in disagreement remained with the users of human-based models (7.6%) (Figure 12).

Regarding the need of a cultural change to take place in the scientific community, answers were less homogeneous: animal-based model users were the least convinced, with the majority of them (60.0%) disagreeing. Users of both animal- and human-based models, and of human models, were more positive in this regard. However, in both cases, significant disagreement was recorded (53.8% positive vs. 33.3% negative for the users of human-based models, and a slight inversion, 40.9% positive vs. 47.7% negative for the users of both types of model) (Figure 12).

A need for concrete regulatory support was also seen as a requirement by the majority of three groups, with 54.9% of the respondents convinced of it (21.2% human-based, 14.2% animal-based and 19.5% both model users). While 26.5% of respondents felt no need for concrete regulatory support to phase out animal research (6.2% human-based, 8.8% animal-based and 11.5% both model users).

The need for wider collaboration between the different stakeholders was seen as a requirement for the phasing out by 48.7% of the contributors, especially from human- and animal-based model users. While 18.8% shared a neutral stance on this point, and 29.1% disagreed.

The need to identify the drawbacks for each field of research for risk analysis was shared by all researchers. Particular agreement on this was recorded among the users of both animal- and human-based models (65.9%). Fifteen users of animal-based models (50.0%), and 20 users of human-based models (51.3%) shared this view. Twenty-eight (24.8%) of all the respondents disagreed, with the higher percentage of them being users of human-based models.

When questioned on the importance of making education on new models and methods mandatory through academic and extra-academic courses, 49.5% of all surveyed responded positively, whilst 36.3% disagreed. By segmenting the votes by type of user, it emerged that the majority (59.0%) of human-based model users considered this as a necessity, followed by 50% of users of both types of model, while only 36.7% of animal-based model users were convinced of the need for this action. On the contrary, the majority of animal-based model users was convinced that this was not a necessary condition to reach the phasing out of animal research (53.3%), compared to 28.2% in the case of human-based model users, and 31.8% for users of both types of model (Figure 12).

## 4. Discussion

During the last decade, the debate on the use of animals in science has gradually intensified. The greater involvement of a variety of different stakeholders has produced a reshaping of the policy of science, and heightened both societal attention and expectations. This enlarged participation [12,21] in one of the key fields of society, biomedical research, is taking place at higher levels where still few health and life sciences researchers are present.

Looking at the animal use statistics, biomedical scientists will be the most affected by policy changes in animal experimentation. However, in our experience there is a large number of biomedical researchers affected by the ongoing animal experimentation debate that are not properly informed or are not properly heard by the relevant stakeholders.

This is the reason why we decided to collect the health and life sciences researchers’ opinions and visions of what the near future holds for biomedical research. This study was meant as a first step towards an understanding of the varying views of the researchers, and a starting point for further investigations to provide policymakers with first-hand information on what the field professionals express as their preferences and expectations.

From a policy-making point of view, and in light of the ongoing debate, we deemed it important to analyse the whole life cycle of experimental models, from its initial selection to its daily use and teaching to the following generations, and the future expectations in the area. This is because a model defines the research activity and embodies the deeper perspective of how research ought to take place, as well as its limits.

Analysing the data, interestingly, we observed a general homogeneity of answers on a multiplicity of topics independently of the segmentation analysis we performed. This observed overarching uniformity may be suggestive of a common “Gestalt” shared by the research community as a whole. Although differences of opinion were expressed by researchers using different models (human-based, animal-based, or both models), our results suggested that different groups of biomedical researchers saw the scientific endeavour through common lenses, which might well provide a needed common ground to establish a fruitful dialogue among researchers who invested in different models. Still, we must acknowledge two specific limitations in this study, which is very important to calibrate further discussion: (1) the number of participants was not statistically representative of the health and life sciences researchers population working in Europe, and (2) 48.7% of our respondents operated in the Netherlands. Even though we performed a statistical control for the sampling bias (see Section 2) between the sampled population from the Netherlands and the rest of participants for certain answers, we cannot exclude further sources of a possible bias in the overall analysis, especially considering the ambition of the Dutch government [30].

Reviewing the collected data, only a minority of respondents declared using a model that was part of their education curriculum. Another minority reported that other models possibly useful in their activity existed, but that they had no access to them. This answer might suggest that model choice is sometimes driven by the level of accessibility inside the research group or environment.

We also looked at the other end of the cycle, which is the selection of models for teaching. This is an aspect of great importance, because through teachers, future researchers are exposed to certain perspectives on research at a key junction of their career, when they are still building their whole perception of the scientific endeavour. In this case, some interesting differences emerged among the three groups of model users. The majority of animal-based model users, and that of users of both types of models, reported presenting their students both animal and human experimental works. However, the majority of human-based model users instead reported teaching their students only through human experimental work. A whole host of interesting considerations on this marked discrepancy between the three groups might be surmised and tested through further surveys, with a view to ascertaining whether it might affect students in their future career [31,32]. Suggestive is also the fact that the development of one’s own teaching tools or media was the most common answer (37.7%) among respondents, followed at some distance (21.1%) by the use of tools/media already established in the field or department.

Surveying the participants on continuing education also emerged interesting results. Human-based model users reported a low rate of attendance to workshops or courses compared to animal-based model users. Furthermore, few human-based researchers attended courses on experimental design improvement, compared to the other two groups. This may be due to a reduced educational portfolio targeting human-based model users. In fact, the knowledge-sharing portfolio on new approach methodologies has only recently been updated and augmented, as shown by the EC-JRC [33].

As to the reasoning behind model choice itself, respondents were adamant on the importance that the chosen model is widespread and established in the scientific community. This overwhelming majority posed high value in working within an established environment with its accepted models and methods, as opposed to performing more exploratory research relying on less (or not at all) acknowledged models or methods. A common view held by the participants was also that the use of non-peer-reviewed models actually hampers the comparison of results among models, with animal-based model users in particular more concerned that the use of such non-peer-reviewed models may challenge supervisors engagement and publications output. However, when subjected to a series of questions about still-non-peer-reviewed models, a majority from all the three groups reported that relying on such models might actually increase the chances of breakthrough solutions. Taken together, these three tendencies would seem to return a somewhat paradoxical view of the scientific endeavour, where science thrives on community-shared models and methods, and scientific activity potentially suffers when straying from this path, and yet, the use of non-peer-reviewed methods (that is, models not yet sanctioned by the community) may actually be the key for breakthrough solutions. This is a quite suggestive outcome. In fact, it could suggest a non-sterile and rather healthy vision of scientific activity, which pursues an out-of-the-box exploration.

Our analysis also highlighted another interesting driver in the selection of experimental models. The majority of the participants stated that animal experiments are a necessary complement to in vitro/in silico human-based experiments, to publish in high impact journals or for business exploitation. This outcome may imply the existence of a discrepancy between research activity and its interpretation by editors and the business sector. In particular, users of animal-based models and of both animal- and human-based models declared direct experience of this situation, which was more nuanced for users of human-based models. This result should elicit a series of questions on where exactly the connection between bench and editorial scientists fails. In the case of business exploitation, translational biomedical research is regulated by the European Medicines Agency (EMA) guidelines, which usually require experimentation in two different animal species. Hence, the pharmaceutical and biomedical sector must provide such evidence, driving translational biomedical researchers to animal-based models. However, the EMA is moving also toward the implementation of the 3Rs for regulatory testing of medicinal products [34], which can represent another medium-term change in biomedical research.

As previously stated, the dialogue on biomedical research is growing in participation, and in consequence. For this reason, the opinion of health and life sciences researchers regarding the participation of several stakeholders in the science policy discussion carries significant interest. Despite some differences, all three user groups (human, animal, both) had a common view on all the stakeholders, with a positive outlook on almost all of them, suggesting the respondents hold a positive view of this enlarged participation. Of note, all the three groups also agreed on considering as dangerous only a specific stakeholder group, political parties. This is in itself a surprising result, which might merit further investigation in itself, considering the weight political parties carry in translating societal calls into policy.

Some very interesting data emerged when the respondents were asked about their outlook on the future of research: a) the majority expected an increase in the scrutiny of research methods/models in grant applications; b) they also expected that the effective benefits and impacts will carry a growing weight; c) lastly, there was a general consensus that national funding will increase for non-animal-based methods, but not for animal-based models/methods. Almost independently of the experimental model used, the participants seemed to be clearly aware of a tectonic shift taking place, not yet fully developed, but certainly expected to loom on the horizon.

The researchers’ outlook on the policy of science led us straight to one of the cornerstones of the entire European health and life sciences research: Directive 2010/63/EU on the protection of animals used for scientific purposes [2]. We proposed to the participants a convoluted question to register their personal views of the Directive itself, asking their opinion on if, how and to what extent the Directive acknowledges animal research in the face of equally reliable alternative methods. A majority of the respondents answered that they do not know—an answer for which the authors cannot fully rule out the complex structure of the question. However, a reasonable number of respondents appeared convinced that the Directive is not objective toward animal research, because its ultimate aim is to phase out the use of animals in science. This is certainly a result worthy of attention, especially in view of the ongoing debate on the topic, both within the scientific community and externally, and its growing polarization [35,36,37,38].

The growing societal attention to the use of animals in science, and on what conditions might lead to a step-by-step replacement of their use, has been subjected to further investigation. A general agreement was registered among the three groups of users on the need to “fill the knowledge gap” before the use of animals can be made unnecessary, as the research community still felt the existence of unsatisfactory areas which need improvement. Researchers using human-based models also saw a need for a cultural change in the scientific community, an opinion that was not completely shared by animal-based model users. In addition, this latter group did not believe in making courses on new models/methods compulsory as academic and extra-academic courses, whereas those researchers also using human-based models considered this another important measure to move the field forward. It is also important to highlight that the majority of researchers agreed on the need to perform an accurate analysis of drawbacks and potential risks for each sector, before a transition away from animals can be finalized. This call by researchers to perform a specific analysis for each sector of health and life sciences points out the presence of unignorable differences in the perception of the state of the art of each branch of the health and life sciences, which can hinder the transition to a non-animal experimental research. On this specific issue, the reports in different disease areas on human-based methods and models, published by the EC-JRC, have shown a large difference in the proportion of use of human-based models depending on the biomedical research field [39,40,41,42], reinforcing with evidence the perceived differences in the state of the art of the biomedical research fields in implementing human-based models/methods we registered in this study.

In conclusion, notwithstanding the modest cohort of researchers that participated in our survey, this study established solid pillars on which future investigations can build to appraise specific social aspects of the health and life sciences research community. We invite other researchers and stakeholders to avail themselves of the analyses and conclusions contained in this study as a possible means to bridge some of the critical communicational and interpretational gaps between the scientific community and society at large, so as to better frame and further the essential debate on the governance of health and life sciences research [43]. The communication gap between the professionals of science and the other interested parties is certainly playing a role in hampering a healthier and more inclusive debate on the future of biomedical research, through an often unspoken yet subtly detrimental bias.

## 5. Conclusions

Researchers with very different backgrounds and perspectives did show homogeneous opinions on a variety of surveyed key topics. This underlying similarity could be used to promote dialogue between groups that may feel having little in common, helping them to converge. The majority of respondents chose their models among those already accepted by the research community and published, making collaborations and results comparison easier. However, researchers also seemed convinced that non-peer-reviewed models have the greater chances to produce breakthroughs. The models effectively used, very rarely, were part of the educational curriculum of a researcher, suggesting a fracture between education and professional activity. The majority of researchers shared the view that human-based in vitro/in silico models must be complemented by animal experiments to secure high-impact factor scientific publication and for business exploitation. The majority of them reported having had direct experience of this. Researchers seemed convinced that the majority of stakeholders intervening in the science governance are actually beneficial, with one exception. Political parties were considered as a dangerous stakeholder group by the majority of researchers, which is an interesting result deserving further research, considering the effective weight political parties carry in translating societal calls into policy. Researchers were convinced that scrutiny will increase for research proposals in terms of models used, and for potential benefits and impacts are expected. Furthermore, they expected a funding increase for non-animal methods in the mid-term. A number of researchers considered that the Directive is not objective towards animal research, as it aims to its gradual stop. This finding is worthy of attention because it highlights that more efforts are needed to ensure researchers fully understand the Directive 2010/63/EU itself. The majority of researchers believed there is still a knowledge gap to be filled, and appropriate risk/benefit analyses must be performed for each specific field before phasing out animal experimentation. A large majority is convinced further regulatory support is needed.

## Figures and Tables

**Figure 1 animals-12-02778-f001:**
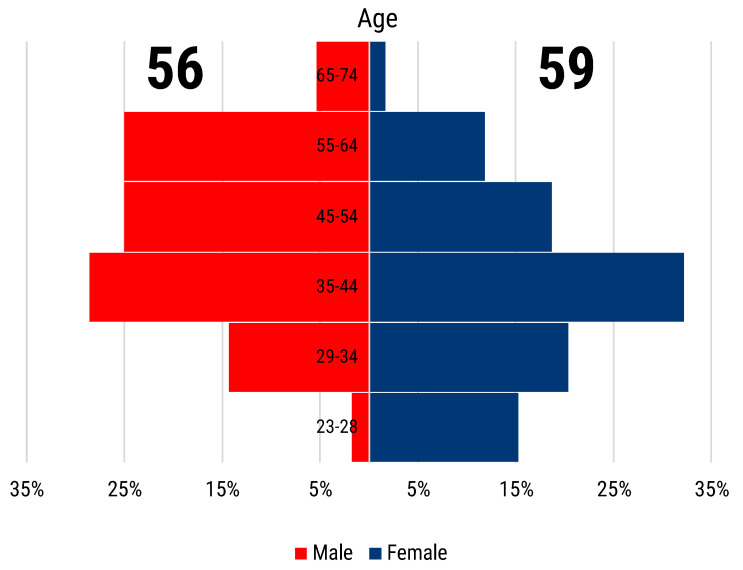
Population pyramid of participants, with 56 males (red) and 59 females (blue) grouped in six age ranges.

**Figure 2 animals-12-02778-f002:**
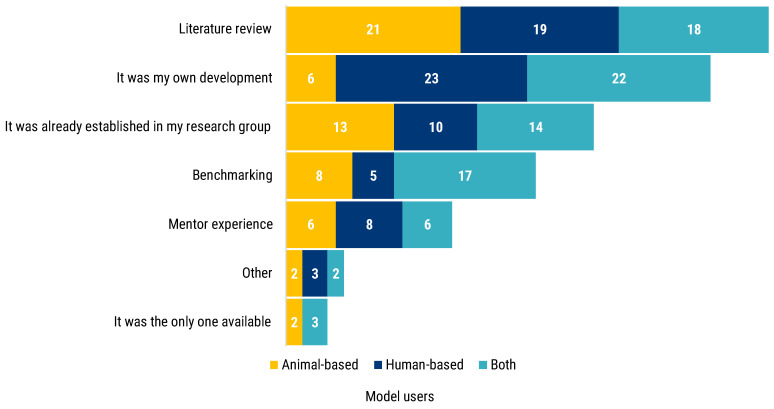
Participants answered the following question “How did you choose or establish your main experimental model?”. The stacked bar chart shows the frequency of each answer segmented by the researcher’s use of animal- (yellow) or human-based (blue) or both models (turquoise).

**Figure 3 animals-12-02778-f003:**
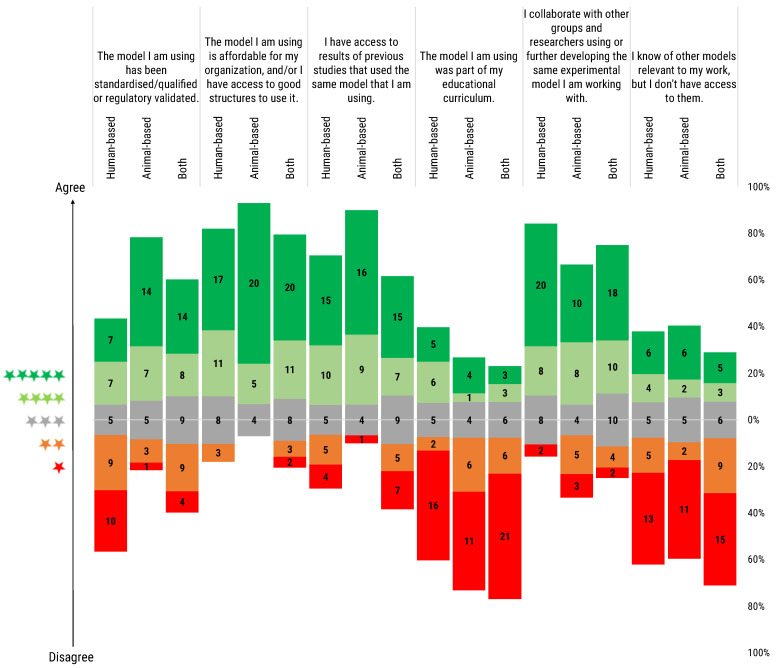
Participants were asked to rate the following statements regarding the use of models. Rating scale was from 1 (red) to 5 (green) stars. Histogram dimensions are represented as 100%, where 100% is the total number of participants of each subgroup: human-based (39), animal-based (30) and both (44) model users. The stacked chart, calibrated around the neutral score (3 stars), shows the number of respondents based on their degree of agreement and segmented by researchers’ use of animal- or human-based or both models.

**Figure 4 animals-12-02778-f004:**
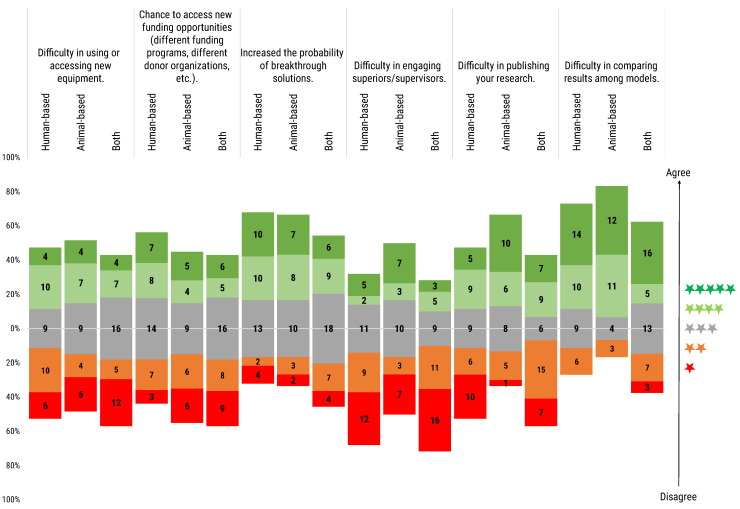
Participants were asked to rate the answers to the following question regarding concerns and benefits of use non-peer-reviewed models, “If you had the need to use a still-non-peer-reviewed model or method, how would you rate the following concerns and benefits?”. Rating scale was from 1 (red) to 5 (dark green) stars. Histogram dimensions are represented as 100%, where 100% is the total number of participants of each subgroup: human-based (39), animal-based (30) and both (44) model users. The stacked chart, calibrated around the neutral score (3 stars), shows the number of respondents based on their degree of agreement and segmented by researchers’ use of animal- or human-based or both models.

**Figure 5 animals-12-02778-f005:**
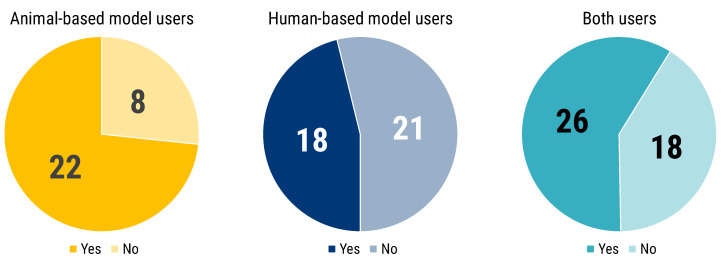
Participants answered the following question, “In your opinion, in order to publish in high-impact factor journals or for business exploitation (industry licensing, etc.) are animal experiments necessary to complement in vitro/in silico human-based experimental models?”. The pie charts show the number of respondents answering “Yes” (blue) and “No” (red) segmented by researchers’ use of animal- (shades of yellow) or human-based (shades of blue) or both models (shades of turquoise).

**Figure 6 animals-12-02778-f006:**
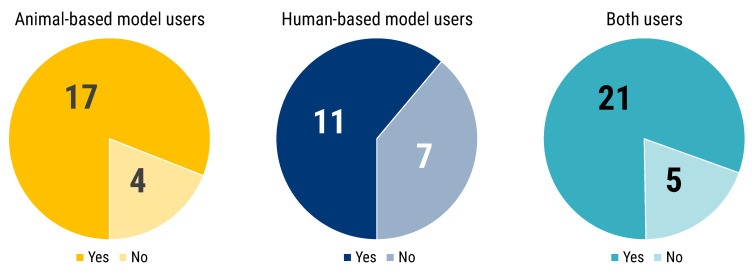
Participants answered the following question “Did you have any direct experience?”. This chart is linked to the previous question (see Figure 5). The pie charts show the number of respondents answering “Yes” (blue) and “No” (red) segmented by researchers’ use of animal- (shades of yellow) or human-based (shades of blue) or both models (shades of turquoise).

**Figure 7 animals-12-02778-f007:**
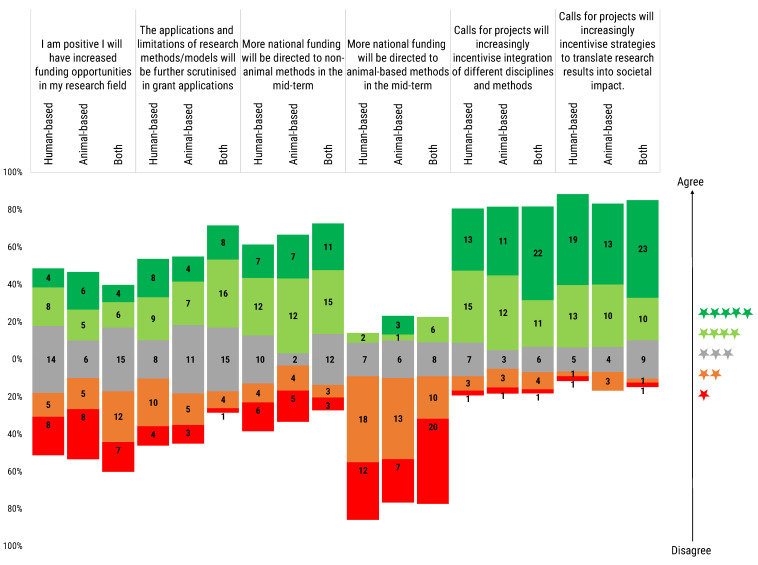
Participants were asked to rate a series of statements regarding funding opportunities. Rating scale was from 1 (red) to 5 (green) stars. Histogram dimensions are represented as 100%, where 100% is the total number of participants of each subgroup: human-based (39), animal-based (30) and both (44) model users. The stacked chart, calibrated around the neutral score (3 stars), shows the number of respondents based on their degree of agreement and segmented by researchers’ use of animal- or human-based or both models.

**Figure 8 animals-12-02778-f008:**
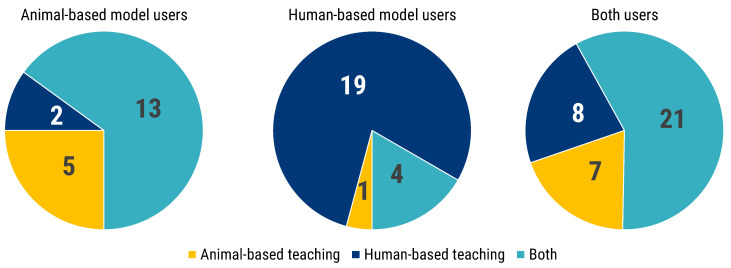
Participants answered the following question “In your teaching activity, are you mostly presenting human or animal experimental works ?”. The pie charts show the number of respondents answering regarding their teaching of experimental work. Each pie represents the answers for each group of researchers (animal-based, human-based or both models users).

**Figure 9 animals-12-02778-f009:**
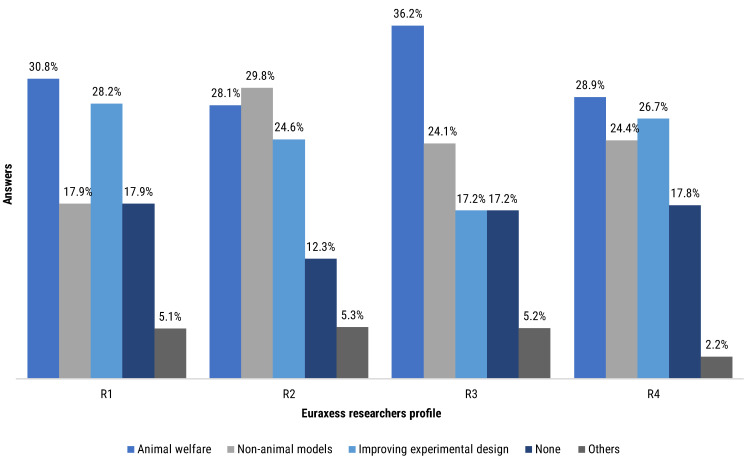
Participants answered the following question “Have you ever attended any workshop/course on …?” by selecting closed options to choose from a list. The bar chart represents the distribution of answers regarding continuing education topic reported by each Euraxess research profile (the bar chart is normalised to the total answers per group: R1 = 39, R2 = 57, R3 = 58, R4 = 45).

**Figure 10 animals-12-02778-f010:**
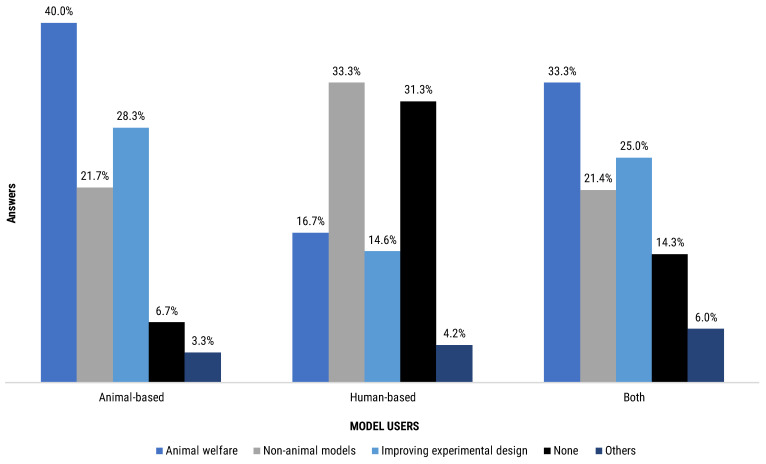
Participants answered the following question “Have you ever attended any workshop/course on…?” by selecting closed options to choose from a list. The bar chart represents the distribution of answers regarding continuing education topic reported by type of mode users (the bar chart is normalised to the total answers per group of model users: animal-based models = 60, human-based models = 48, both models = 84).

**Figure 11 animals-12-02778-f011:**
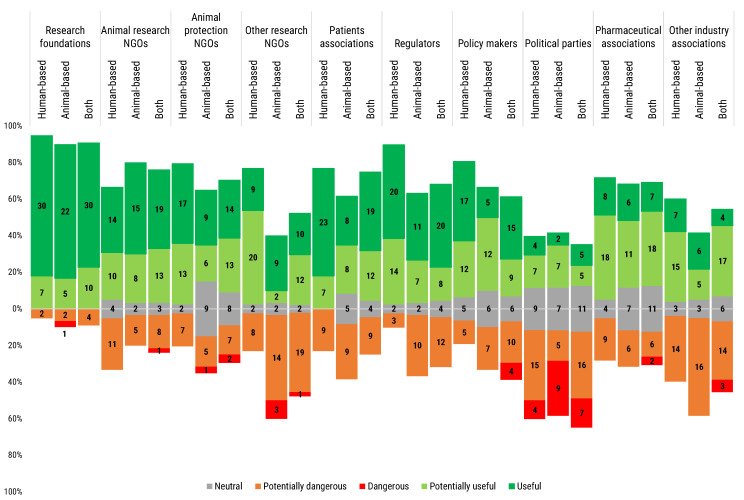
Participants answered the following question “What is your opinion on the participation of the following organisations in the scientific and political debates regarding the orientation of research?”. The stacked chart, calibrated around the neutral answers, shows the number of respondents based on their opinion, from “Dangerous” (in red) to “Useful” (in dark green), and segmented by researchers’ use of animal- or human-based or both models. Histogram dimensions are represented as 100%, where 100% is the total number of participants of each subgroup: human-based (39), animal-based (30) and both (44) model users.

**Figure 12 animals-12-02778-f012:**
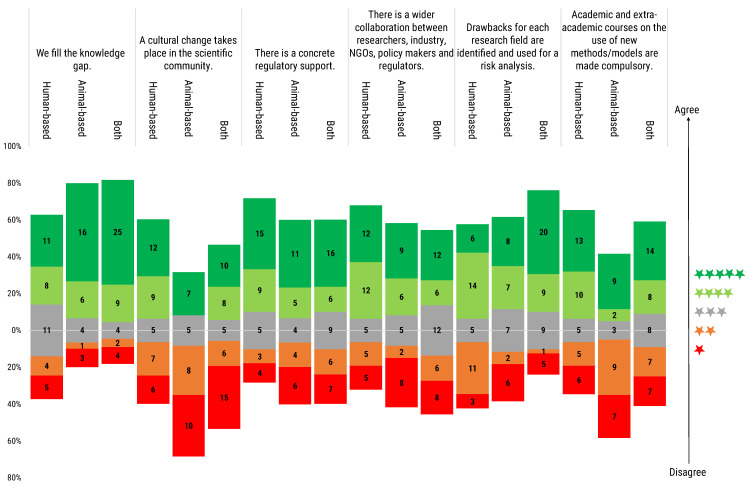
Participants were asked to rate the following statement, “Animal-based research should be phased out only if …”, completed with a set of different options regarding the gaps or barriers to overcome for phasing out animal experimentation. Rating scale was from 1 (red) to 5 (dark green) stars. Histogram dimensions are represented as 100%, where 100% is the total number of participants of each subgroup: human-based (39), animal-based (30) and both (44) model users. The stacked chart, calibrated around the neutral score (in grey, 3 stars), shows the number of respondents based on their degree of agreement and segmented by researchers’ use of animal- or human-based or both models.

**Table 1 animals-12-02778-t001:** Participants answered the following question “If an animal model and a new non-animal model provide comparable results, do you think the Directive 2010/63/EU acknowledges adequately the animal model by virtue of its being well known, predictable, widely used and established?” Data are presented as percentage of participants segmented by Euraxess research profile to provide a normalized view of results view.

Group	R1	R2	R3	R4
Poorly, because its aim is that of phasing out animal research, so it cannot be fully objective.	-	19.4%	18.4%	7.7%
Significantly, because the Directive acknowledges the currently irreplaceable value of animal research.	13.6%	12.9%	7.9%	26.9%
No, because the Directive requires the use of alternative methods and techniques if they are available.	22.7%	29.0%	18.4%	23.1%
I do not know.	63.6%	38.7%	55.3%	42.3%
TOTAL	100% (24)	100% (42)	100% (51)	100% (26)

**Table 2 animals-12-02778-t002:** Participants answered the following question “If an animal model and a new non-animal model provide comparable results, do you think the Directive 2010/63/EU acknowledges adequately the animal model by virtue of its well known, predictable, widely used and established?” Data are presented as percentage of participants segmented by type of model users to provide a normalized view of results view.

Group	Animal-Based Model Users	Human-Based Model Users	Both
Poorly, because its aim is that of phasing out animal research, so it cannot be fully objective.	20.0%	12.8%	9.1%
Significantly, because the Directive acknowledges the currently irreplaceable value of animal research.	10.0%	12.8%	18.2%
No, because the Directive requires the use of alternative methods and techniques if they are available.	16.7%	30.8%	20.4%
I do not know.	53.3%	43.6%	52.3%
TOTAL	100% (30)	100% (39)	100% (44)

## Data Availability

The data presented in this study are openly available in ZENODO at https://doi.org/10.5281/zenodo.7077609.

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
