# Peer review of "Researchers and Their Experimental Models: A Pilot Survey in the Context of the European Union Health and Life Science Research"

_animals, 2022, doi:10.3390/ani12202778_

Round 1

Reviewer 1 Report

Dear Authors,

the content is very interesting but there is the need to significantly improve  the presentation of the results. In detail: Figure 2 should be moved from page 12 line 485 to page 6 line 218. Table 1 should be moved from page 4 line 179 to page 14 line 598 and table 2 from page 6 line 230 to page 14 line 617. Supplementary figures are missed (I cannot find them in the submission).

Further, I made some minor changes in the text ( see the attached doc). 

Author Response

Dear Reviewer 1,

Thanks a lot for your major interest in our study. 

We are sorry for the several formatting errors that were present due to the conversion from our Word format to the journal format. We addressed all the formatting issues regarding the results presentation, as figures location in the text and their numbering. We also clarified the journal formatting, placing the correct figures and table instead of the supplementary ones, which are now placed at the end of the manuscript we are providing now to the editors.

We really appreciate and accept the minor changes suggested in the text suggested, improving text readability.

Please find below the detailed description of the changes applied to the new manuscript version.

Kind regards,

Dr. Marco Straccia

Changes details:

  • Figures locations in the text have been corrected and improved.
  • Figures numbering in the text has been corrected.
  • Supplementary figures were wrongly presented instead of the proper corresponding figures. They have been removed. Correct figures are now placed in the corresponding section of the text and Supplementary figures are presented at the end of the manuscript.
  • Supplementary tables were removed from the text and moved at the end of the manuscript.
  • Proper tables 1 and 2 were placed in the corresponding positions.
  • Figure and Table legends were corrected and improved accordingly.
  • We added a detailed statement in the Methods section and Institutional Review Board Statement section.
  • Three new references have been added accordingly to the insertion of the IRB statement and the Informed consent statements.
  • “Del Pace et al. 2022” text has been removed from all figures.
  • Minor suggested changes have been implemented as suggested by you and Reviewer 2.
  • We added a conclusive statement in the Abstract, as suggested by Reviewer 3.
  • We changed the manuscript’s title, as suggested by Reviewer 3, to better capture the whole information contained in the present study that according to the stated aim of the study, it is not only focused on the replacement of animal models in research. The new proposed title is: “Researchers and their experimental models: A pilot survey in the context of the European Union health and life science research.”
  • An explanatory statement has been added in the text (line 295-297) to clarify the meaning of the categories: animal-based models and human-based models.
  • We improved figure headings readability.

Reviewer 2 Report

The authors of this article Lorenzo Del Pace et al. present a study that involves collecting opinions, through a questionnaire, from researchers from different areas of work.

This is a very interesting topic since nowadays there is a lot of public discussion about the use of animal experimental models.

In my view, it has some limitations, which are even detected and described by the authors throughout the article, namely with regard to the selection of the study population.

Despite the limitations presented, I am of the opinion that the work presents very interesting results, namely in the consensual issue that animal models are necessary to complement in vitro/in silico studies.

In conclusion, I think this article can be accepted for publication, but there are some text editing errors that should be corrected.

Author Response

Dear Reviewer 2,

We really appreciate your major interest in our study. As you said, we fully recognise the limitations of our study and for this reason we fully acknowledge them in the article itself, in order to be improved by future research groups in this area.

We have addressed the text editing issues, such as the formatting, the figures/tables numbering and location in the text. We moved the supplementary figures and table at the end of the text, placing the right ones in the corresponding section of the manuscript. We also simplified the text to improve readability where needed.

Please find below the detailed description of the changes applied to the new manuscript version.

Kind regards,

Dr. Marco Straccia

Changes details:

  • Figures locations in the text have been corrected and improved.
  • Figures numbering in the text has been corrected.
  • Supplementary figures were wrongly presented instead of the proper corresponding figures. They have been removed. Correct figures are now placed in the corresponding section of the text and Supplementary figures are presented at the end of the manuscript.
  • Supplementary tables were removed from the text and moved at the end of the manuscript.
  • Proper tables 1 and 2 were placed in the corresponding positions.
  • Figure and Table legends were corrected and improved accordingly.
  • We added a detailed statement in the Methods section and Institutional Review Board Statement section.
  • Three new references have been added accordingly to the insertion of the IRB statement and the Informed consent statements.
  • “Del Pace et al. 2022” text has been removed from all figures.
  • Minor suggested changes have been implemented as suggested by you and Reviewer 2.
  • We added a conclusive statement in the Abstract, as suggested by Reviewer 3.
  • We changed the manuscript’s title, as suggested by Reviewer 3, to better capture the whole information contained in the present study that according to the stated aim of the study, it is not only focused on the replacement of animal models in research. The new proposed title is: “Researchers and their experimental models: A pilot survey in the context of the European Union health and life science research.”
  • An explanatory statement has been added in the text (line 295-297) to clarify the meaning of the categories: animal-based models and human-based models.
  • We improved figure headings readability.

Author Response

Dear Reviewer 3,

Kind Regards,

Dr. Marco Straccia

Round 2

Reviewer 3 Report

The authors have made significant changes in the manuscript. I am happy to approve it for publishing.